# Mesenchymal Stem Cells–Hydrogel Microspheres System for Bone Regeneration in Calvarial Defects

**DOI:** 10.3390/gels8050275

**Published:** 2022-04-29

**Authors:** Chong Teng, Zhicheng Tong, Qiulin He, Huangrong Zhu, Lu Wang, Xianzhu Zhang, Wei Wei

**Affiliations:** 1Department of Orthopaedic Surgery, The Fourth Affiliated Hospital, Zhejiang University School of Medicine, Yiwu 32200, China; tengchong1984@zju.edu.cn (C.T.); tongzc@zju.edu.cn (Z.T.); zhrsmu@163.com (H.Z.); 2Key Laboratory of Tissue Engineering and Regenerative Medicine of Zhejiang Province, Zhejiang University School of Medicine, Hangzhou 310058, China; 11971014@zju.edu.cn; 3Dr. Li Dak Sum & Yip Yio Chin Center for Stem Cells and Regenerative Medicine, and Department of Orthopedic Surgery of the Second Affiliated Hospital, Zhejiang University School of Medicine, Hangzhou 310058, China; 4Department of Pathology, The Fourth Affiliated Hospital, Zhejiang University School of Medicine, Yiwu 32200, China; wl0413@zju.edu.cn

**Keywords:** calvarial defect, bone marrow mesenchymal stem cells, hydrogel microspheres, bone regeneration

## Abstract

The repair of large bone defects in clinic is a challenge and urgently needs to be solved. Tissue engineering is a promising therapeutic strategy for bone defect repair. In this study, hydrogel microspheres (HMs) were fabricated to act as carriers for bone marrow mesenchymal stem cells (BMSCs) to adhere and proliferate. The HMs were produced by a microfluidic system based on light-induced gelatin of gelatin methacrylate (GelMA). The HMs were demonstrated to be biocompatible and non-cytotoxic to stem cells. More importantly, the HMs promoted the osteogenic differentiation of stem cells. In vivo, the ability of bone regeneration was studied by way of implanting a BMSC/HM system in the cranial defect of rats for 8 weeks. The results confirmed that the BMSC/HM system can induce superior bone regeneration compared with both the HMs alone group and the untreated control group. This study provides a simple and effective research idea for bone defect repair, and the subsequent optimization study of HMs will provide a carrier material with application prospects for tissue engineering in the future.

## 1. Introduction

Bone tissue has the ability to repair itself. Most fractures can heal well after a few months. However, in some cases, this ability to repair itself fails. For example, large bone defects, caused by tumors or trauma, cannot be healed by bone tissue itself [1,2]. The reason is that the hollow structure of large bone defects cannot provide a medium of sufficient mechanical strength to support bone tissue repair. Therefore, a bone graft is the most commonly used therapy in clinical practice, which can provide a strong support structure [3,4]. Bone grafts can be divided into autograft and allograft. Autografts require surgical removal of healthy bone tissue of appropriate size, which can easily cause a second blow to the patient’s body. Allografts require facing the risk of immune responses and cross-infection, which severely limits their application. In addition, as most bone defects are irregular, bone graft materials are not effective in filling bone defects and cause unnecessary waste [5,6,7].

Recently, tissue engineering has become a promising solution for bone defect repair [8,9]. Tissue engineering is the technology of regenerating corresponding tissues or organs by combining materials and cells, which consist of scaffold, seed cells, and growth factors. The ideal bone tissue scaffold material needs to have the following characteristics: biocompatibility, osteoinductive activity, suitable aperture ratio and aperture size, sufficient mechanical strength, and plasticity. The biggest challenge is that the ideal tissue engineering scaffold material cannot balance mechanical strength and plasticity [10]. This means that tissue engineering scaffolds cannot fill well with irregular bone defects.

Hydrogel is a kind of three-dimensional network structure with excellent plasticity [11]. Studies have shown that hydrogels can be used to repair bone defects [12,13]. Cells can also be cultured in a hydrogel system to promote bone formation [14]. However, this hydrogel system has disadvantages of poor cell nutrient exchange and insufficient mechanical strength [15]. Recently, hydrogel microspheres (HMs) were developed to solve the problems of traditional hydrogels, such as large external size, long cell culture cycle, and the need for surgical implantation of diseased or necrotic areas [16]. Studies have shown that HMs can not only be used for minimally invasive delivery of biological agents, but also can be used to form microporous scaffolds to promote cell growth and adhesion, and can also be embedded into large hydrogels to construct multilevel structural materials [17]. Therefore, HMs have promising applications in biomedical fields such as delivering therapeutic drugs, constructing tissue repair scaffolds, and bio-inks for 3D printing. In the field of tissue engineering for bone defect repair, HMs are increasingly showing their advantages, especially that which He et al. reported: an “All-in-One” HMs system for stem cell amplification which is promising for tissue engineering [18]. At present, researchers are exploring the use of HMs as cell carriers to repair bone and cartilage tissues, and have widely explored the use of HMs as cell carriers to promote heart repair and the use of HMs to carry islet cells to treat diabetes [19,20,21,22].

Here, in this study, we aim to develop a stem cell/HMs system as a strategy to address the issues of bone regeneration in calvarial defects (Figure 1). Firstly, a suitable size HM was prepared by a microfluidic system based on light-induced gelatin of GelMA. Secondly, bone marrow mesenchymal stem cells (BMSCs) were cultured on the surface of HMs by taking advantage of gelatin to promote cell adhesion. Finally, we assembled the stem cell/HMs system which was then applied to a rat skull defect. Micro-CT and histological examination were used to evaluate the effect of the bone defect repair.

## 2. Results

### 2.1. Physical and Biocompatible Characterization of HMs

In this study, HMs were fabricated by microfluidic technology which could produce microspheres with uniform size [23]. According to the microscopic photograph, we found that HMs are transparent regular spheroids (Figure 2A,B). The diameter of the microspheres is about 282.92 ± 3.82 µm. Additionally, it has a mechanical property of about 2.28 kilopascals (kPa) determined by a Nanoindenter (Figure 2C). This means that HMs have sufficient mechanical strength to support bone repair and to cope with a possible increase in intracranial pressure, which is normally between 0.69 and 1.96 kPa.

Next, we evaluated the biocompatibility of HMs. BMSCs from the femur and tibia of rats were co-cultured with HMs. In the control group, the same density of stem cells was seeded on normal cell culture plates. After 1, 4, and 7 days, live/dead staining was carried out, respectively, to analyze the cytocompatibility of HMs. As shown in Figure 2D, the viability of BMSCs co-cultured with HMs for 1, 4, and 7 days is unaffected. Obviously, there are few red dead cells, while green living cells are clearly visible. Interestingly, we found that the longer the culture time, the more cells adhered to the surface of the HMs. This may be related to the retention of RGD sequence in GelMA to promote cell adhesion. Semi-quantitative analysis of the live/dead staining plots showed that the cell survival rate remained at a high level of about 99% (Figure 2E). Moreover, the CCK-8 assay indicates that there is no significant difference in cell proliferation between the control group and BMSC/HMs co-cultured group (Figure 2F). Meanwhile, the HMs did not show cytotoxicity with the increase in culture time (Figure 2G).

### 2.2. Differentiation Effect of HMs on Stem Cells

It is well known that stem cells are the main source of osteoblasts, so we preliminarily analyzed the effect of HMs on stem cell differentiation. In this study, BMSCs were cultured at a density of 3× 10^5^ cells/disc in a 6-well plate. In the HMs group, to each well 330 μL medium containing microspheres was added. The control group was culture with normal medium without HMs. After 21 days of co-culture, we detected the gene expression levels of osteogenic, chondrogenic, and adipogenic differentiation of stem cells by qRT-PCR. As shown in Figure 3, three osteogenic marker genes were significantly increased while the expression of adipogenic differentiation marker genes did not change. The effect of HMs on chondroblast differentiation showed the opposite result, which promoted SOX9 expression and inhibited the level of COLL2. However, the effect of HMs on chondrogenic differentiation is very limited because of the small multiple changes in marker genes. In general, HMs showed a good effect on promoting bone differentiation, which indicated that the co-culture of stem cells and HMs had the ability to repair bone defects.

### 2.3. In Vivo Evaluation of BMSC/HMs System for Bone Regeneration with Skull Defect Model

To characterize the osteogenic effect of BMSC/HMs in vivo, we first constructed a rat skull defect model with a dental drill. A circular defect was created in the skull with an 8mm diameter dental drill. Additionally, after 7 days of co-culture, BMSC/HMs were then implanted into the defect area with hydrogel as the medium. The microspheres implantation group and the blank group were used as the control groups.

After 8 weeks, all rats were killed to obtain skull specimens. Additionally, micro-CT was then utilized to observe the formed bone tissue at the defect area. The results show that more regenerated bone tissues formed in the BMSC/HMs group as shown in Figure 4A. Based on the micro-CT images, there was obvious new bone formation at the edge of the bone defect in the BMSC/HMs group. In contrast, no obvious regenerated bone formation was observed in the control and HM groups. Consistent with this, bone histomorphometric analysis showed that the bone mineral density of the BMSC/HMs treated group was significantly higher than that of the other two groups (Figure 4B). The BV and BV/TV also showed the same trend. The Tb.Sp of BMSC/HMs treated group, HMs alone treated group, and the untreated control group in the defect area were 0.21 ± 0.02 cm, 0.45 ± 0.03 cm, and 0.38 ± 0.04 cm, respectively. This also indicated that BMSC/HMs promoted bone formation. Furthermore, the Tb.Th and BS/BV were used to evaluate the quality of the newly formed bone. The two data showed no difference, indicating that the regeneration of new bone tissue was uniform. Interestingly, all of these data described a significant osteogenic effect in the experimental group, while microspheres implantation alone did not effectively promote bone mass growth. These results indicated that the existence of stem cells was the key factor to promote bone defect repair. The co-culture of HMs and stem cells plays a synergistic role in bone defect repair and magnifies the therapeutic effect.

Histological staining was used to further assess bone regeneration in the defect area. As shown in Figure 5A, H&E staining showed that the control group and the HMs group still have a little continuity of soft tissue in the bone defect area, while in the BMSC/HMs group there is abundant regenerated tissue. Subsequently, IHC staining of OCN, one of the important osteogenic markers, was used to evaluate newborn bone tissue. From the immunohistochemical image of OCN, there were almost no positive staining cells in the control group and microspheres group. On the contrary, in the BMSC/HMs group, a significantly higher number of OCN-positive cells were found as displayed in the brown color (Figure 5B). Semi-quantitative analysis also showed that after 8 weeks of surgery, the percentages of OCN-positive cells in the control group, HMs alone group, and BMSC/HMs group were 7.41 ± 0.63%, 10.29 ± 1.24%, and 32.18 ± 1.14%, respectively (Figure 5C). Together, these results suggest that both BMSCs and HMs are indispensable, and only when they are combined can they play a significant role in repairing bone defects.

## 3. Discussion

Bone regeneration has always been a difficult problem to be solved in clinic, especially large bone defects caused by tumors, trauma, and other factors, which often lead to poor prognosis and poor quality of life [24]. The current clinical treatment strategy is autologous and allograft. Autologous bone transplantation can damage the patient to a certain extent and allograft is limited by insufficiency and immune rejection [5,6]. In fact, there is another option—artificial bone grafts. However, studies have shown that artificial bone may interfere with blood flow and cause mechanical imbalances, leading to brittleness and deformation of surrounding tissues [25]. In recent years, the rise in tissue engineering has provided a new idea for bone reconstruction. Various tissue-engineering materials emerge endlessly, such as polymer materials, inorganic materials, and metal materials, etc. [8]. However, the pore ratio of tissue-engineered materials is difficult to control. The high porosity ratio promotes the growth of cells, but also weakens the strength of materials, while the low porosity ratio prevents cells and blood vessels from entering the pores [26].

Here, we provide only a simple and effective adhesive material for seeding cells. Cells can freely adhere to the microspheres’ surface, and at the same time realize the three-dimensional culture of cells. It has been reported that the three-dimensional spatial structure of HMs can provide a biomimetic three-dimensional microenvironment, and 3D culture is more conducive to maintaining the stemness, function, and phenotype of stem cells than the two-dimensional amplification culture using cell culture plates [27]. At the same time, its small size and large specific surface area significantly increase the area of cell adhesion growth [28]. More importantly, the material itself is a biodegradable natural substance, which will not cause other negative effects and even promotes cell adhesion through RGD sequences. Our study shows that HMs can promote osteogenic differentiation of stem cells. Additionally, the combination of HMs and BMSCs can significantly promote bone formation.

In fact, GelMA has a broad application prospect in the field of bone defect repair materials due to its controllable degradation performance, controllable mechanical properties, and good biocompatibility. A large number of studies have shown that GelMA’s unique photocrosslinking properties can be used to process hydrogel scaffold materials with different morphologies [29]. At the same time, GelMA hydrogel can meet the diverse research of bone tissue engineering application materials by adding various chemical groups or different synthesis processes. Consistent with our results, GelMA itself promotes the osteogenic differentiation of BMSCs. For example, the concentration of GelMA hydrogels can affect the osteogenic differentiation of hydrogel. 5% GelMA hydrogels have better bone regeneration performance in vitro than 10% GelMA hydrogels [30]. This also explains why, in our in vivo results, BMSC/HMs group significantly promoted bone formation, while the microsphere implantation alone did not differ from the blank group. Stem cells play a key role in bone defect repair [31,32]. Large skull defects, resulting in a lack of cells in the central area of the defect, and damage to surrounding tissue during surgery, negates GelMA’s ability to promote osteogenic differentiation. Furthermore, GelMA HMs were developed to solve the problems of traditional hydrogels, such as large external size, long cell culture cycle, and the need for surgical implantation of diseased or necrotic areas. Their small size enables injection through small needles and catheters and inhalation of particles. Granular hydrogels can possess significant porosity owing to the interstitial space between HMs, which support cell proliferation and migration.

In addition, based on the results of this study, it may be possible to enhance the bone regeneration of the hydrogel by doping different components in future research. Hydroxyapatite (HAP), for example, is the important inorganic component of human bone tissue [33,34]. It can dissolve and release some harmless ions in the body and can participate in metabolism in the body. More importantly, it can induce bone tissue hyperplasia and promote bone defect repair [35]. Therefore, GelMA hydrogels doped with hydroxyapatite can not only improve the ability of bone defect repair, but also enhance the gel strength [36,37]. Others, such as the insertion of octacalcium phosphate in GelMA hydrogel, significantly promoted the differentiation of mesenchymal stem cells into osteoblasts [38]. Other studies have shown that GelMA co-crosslinked with a photocrosslinked osteogenic growth peptide (OGP) can generate GelMA-C-OGP. Its mechanical compression performance can reach 90 kPa, which is superior to GelMA. In addition, GelMA-C-OGP can degrade and facilitate the release of OGP [39]. Besides, blood vessel production is also the key to bone defect repair. Bone is a highly vascularized tissue, and inadequate vascularization often leads to poor bone regeneration. Achieving early vascularization to provide nutrients and oxygen in large bone grafts remains a challenge [40,41]. The use of GelMA hydrogel containing different concentrations of vascular endothelial growth factor can promote microvascular formation [42,43].

Although GelMA hydrogel has properties conducive to osteogenesis such as cell adhesion, proliferation, and differentiation, its mechanical properties limit its application as a bone defect repair material. Our results showed that the BMSC/HMs group promoted bone formation from the periphery of the defect, while no significant bone regeneration was observed in the central area of the defect with weak mechanical strength. This may be because stem cells are mechanical-signal-sensing cells [44,45,46], and HMs lack the corresponding mechanical strength in the defect center. Moreover, there could be more endogenous MSCs around the defect site than those of the defect center. These endogenous MSCs could migrate to the HMs at the periphery of the defect. Studies have shown that MSCs are very sensitive to the stiffness of hydrogel. The higher the elastic modulus of hydrogel, the more inclined MSCs are to osteogenic differentiation. When hydrogels or extracellular matrix lack mechanical strength, MSCs mainly differentiate into adipocytes [47,48]. Integrin, YAP/TAZ, and MAPK signaling pathways may be involved in the mechanism of inducing MSC differentiation [48,49]. Therefore, the implanted MSCs–HMs system can be adhered around the skull defect, with better mechanical support and higher mechanical strength compared with the central area of the defect. This makes the MSCs–HMs system have stronger osteogenic effect around the skull defect. It also indicates that the mechanical strength of hydrogel plays an important role in promoting bone defect repair. Adding rigid materials is one way to improve the mechanical strength of hydrogels. A novel nanocomposite hydrogel composed of GelMA and nanodiamonds not only controls the osteogenesis of MSCs, but also enhances the mechanical properties of GelMA hydrogel [50]. In future studies, we may be able to improve the properties of hydrogels by adding a variety of ingredients to maximize bone regeneration.

## 4. Conclusions

In summary, HMs were successfully fabricated and were demonstrated with excellent biocompatibility. The results showed that the HMs did not cause any toxicity to BMSCs, and could promote their osteogenic differentiation. The rat skull defect was used as the animal model to investigate the role of BMSC/HMs in bone regeneration in vivo. The micro-CT, H&E staining, and OCN IHC staining results showed that BMSC/HMs had a significant effect on bone defect repair, and stem cells were an indispensable factor for bone regeneration.

## 5. Materials and Methods

### 5.1. Preparation of HMs

The basic principle of preparing GelMA HMs is to carry incompatible liquid through different microchannels, generate droplets at the intersection, and then cross-link the droplets to form hydrogel microspheres. Briefly, we used paraffin oil as the continuous oil phase and phosphate-buffered saline containing GelMA with the addition of 0.5% photo-initiator Lithium phenyl-2,4,6-trimethylbenzoylphosphinate (LAP) as the aqueous phase. The two phases were then injected into a syringe and pumped into different microchannels through the syringe. Then, the water phase and continuous phase were injected into the microfluidic device at different speeds by adjusting the injection pump. The prepared hydrogel droplets were further solidified by photocrosslinking for 20 s under UV irradiation at 365 nm wavelength. Next, the oil and active agent on the surface of HMs were removed by repeated washing with acetone and 75% ethanol, and then purified by washing with PBS for 24 h. Finally, the HMs were observed under a microscope.

### 5.2. Cell Culture

Bone marrow mesenchymal stem cells were derived from bone marrow lumens of femur and tibia of rats. Experimental animals were provided by Experimental Animal Center of Zhejiang University. In simple terms, the femur and tibia of the rat were aseptically severed and a certain amount of culture medium was aspirated with a syringe to flush out the cells in the bone marrow cavity. Then, BMSCs were purified and amplified in complete medium. The medium was first half-changed on day 3, then updated every 2 days, and cells were trypsinized and subcultured at 70–80% confluence. All cells used in this study were between passages 3 and 5.

### 5.3. Cell Viability

Cellular viability of BMSCs, co-cultured with HMs, was tested by live/dead cell staining kit (Life Tech, New York, NY, USA). The BMSCs were co-cultured with HMs at a density of 3 × 10^5^ cells mL^−1^. Then, the BMSC/HMs suspension was gently blown and mixed, placed in a 15mL centrifuge tube, and placed in humidified incubator containing 5% CO_2_ at 37 °C. After culturing for 1, 4, and 7 days, BMSCs were first immobilized, then stained with live/dead cell staining dye and imaged with a fluorescence microscope.

### 5.4. Cell Proliferation

BMSCs between passages 3–5 were cultured with HMs at a density of 1 × 10^4^ cells/disc in a 96-well plate (Corning, New York, NY, USA). In the HMs group, 33ul medium containing microspheres was added to each well, and then complete medium was added to 200 µL. In the control group, the density of stem cells was also 1× 10^4^ cells/disc with 200 µL complete medium. The proliferation of BMSCs was evaluated by the CCK-8 reagent according to the instruction. At 1, 3, 5, and 7 days, the medium was replaced with 180 μL of fresh complete medium and 20 μL of CCK-8 reagent (Dojindo, Kumamoto, Japan) in each well. After incubating for 2 h at 37 °C with 5% CO_2_, a spectrophotometer was used to measure the absorbance of the solution at 450nm.

### 5.5. Quantitative Real-Time PCR

BMSCs were cultured at a density of 3 × 10^5^ cells/disc in a 6-well plate (Corning, USA). In the HMs group, into each well 330 ul medium containing microspheres was added. The control group was culture with normal medium without HMs. The final volume of culture medium was 2ml. Half of the microspheres-free medium was exchanged every two days. After 21 days of culture, the quantitative real-time PCR was utilized to evaluate the expression of osteogenic marker genes (RUNX2, OCN, BMP2), chondrogenic marker genes (SOX9, COLL2), and adipogenic marker genes (PPARγ, LPL, C/EBPα) in cells co-cultured with HMs. Total mRNA was extracted from cells by trizol, and then samples were reverse-transcribed into cDNA using Prime Script RT reagent kit (Takara Bio Inc., Shiga, Japan). Finally, SYBR^®^ Premix Ex Taq™ (Takara) was used to detect cDNA. Data were calculated using the comparison Ct (2^-ΔΔCt^) method and standardized to GAPDH. Primer sequences used are listed in Table 1.

### 5.6. Animal Model of Cranial Bone Defect

All SD rats used in this study were from the Animal Research Center of Zhejiang University. The animal experiment in this study passed the ethical review, numbered ZJU20210229. Three groups were set up (*n* = 3): untreated control group (Ctrl), HMs alone treated group, and BMSC/HMs treated group. Rats were injected intraperitoneally with 1% pentobarbital sodium solution at a dose of 30 mg/kg for preoperative anesthesia. Dissection to the skull surface of rats: Then, the cranial bone defect model with diameter of 8 mm was established with dental drill. The sterilized excipients, HMs or BMSC/HMs were implanted into the calvarial bone defects separately and scalp was sutured. After 8 weeks, rat skulls were taken and fixed in 4% paraformaldehyde solution.

### 5.7. Micro-CT

After 8 weeks postoperatively, the fixed skulls were analyzed via the micro-CT system (Sky Scan 1174, Bruker, Billerica, MA, USA). Auxiliary software was used to generate a 3D reconstruction of the selected area. The bone mineral density (BMD), bone volume (BV), bone surface area (BS), percentage of new bone volume against tissue volume (BV/TV), percentage of new bone surface against bone volume (BS/BV), trabecular space (Tb. Sp), and the trabecular thickness (Tb. Th) were calculated.

### 5.8. Histological Analysis

After 8 weeks, skull samples with soft tissue removed were obtained and immersed in 4% paraformaldehyde solution for 2 days. The skull samples were then decalcified in a 10% EDTA solution for 6 weeks. The needle was used to determine the hardness of the bone. The skull samples were then dehydrated with different concentrations of ethanol solutions. Then, samples were embedded in paraffin and sectioned. Finally, staining of hematoxylin–eosin (H&E) was carried to characterize the bone tissue structure.

### 5.9. Immunohistochemistry (IHC)

Osteocalcin (OCN) is a marker of new bone formation secreted by osteoblasts. Three sections of each group were randomly selected for OCN immunohistochemical staining. In simple terms, sections were incubated overnight with anti-OCN primary antibody (1: 100) at 4 °C. Then, the secondary antibody (1:500) was added and incubated at room temperature for 2 h. After the staining was completed, light microscopy was used to observe the three randomly selected representative regions for semi-quantitative analysis of OCN-positive staining. Image Pro Plus 6.0 software (Media Cybernetics) was used to measure the integral optical density (IOD) of OCN-positive staining area. The intensity of each image was calculated by normalizing the IOD with the randomly selected areas of interest.

Osteocalcin (OCN) is a marker of new bone formation secreted by osteoblasts. Three sections of each group were randomly selected for OCN immunohistochemical staining. In simple terms, sections were incubated overnight with anti-OCN primary antibody (1: 100) at 4 °C. Then, the secondary antibody (1:500) was added and incubated at room temperature for 2 h. After the staining was completed, light microscopy was used to observe the three randomly selected representative regions for semi-quantitative analysis of OCN-positive staining. Image Pro Plus 6.0 software (Media Cybernetics) was used to measure the integral optical density (IOD) of OCN-positive staining area. The intensity of each image was calculated by normalizing the IOD with the randomly selected areas of interest.

### 5.10. Statistical Analysis

All data are shown as averaged ± standard deviation (SD). All experiments were performed with 3 biological replicates. The differences between the two groups were statistically analyzed using the Student’s *t*-test, and *p* < 0.05 was considered statistically significant. * indicates *p* < 0.05, ** indicates *p* < 0.01, *** *p* < 0.001, **** *p* < 0.0001.

## Figures and Tables

**Figure 1 gels-08-00275-f001:**
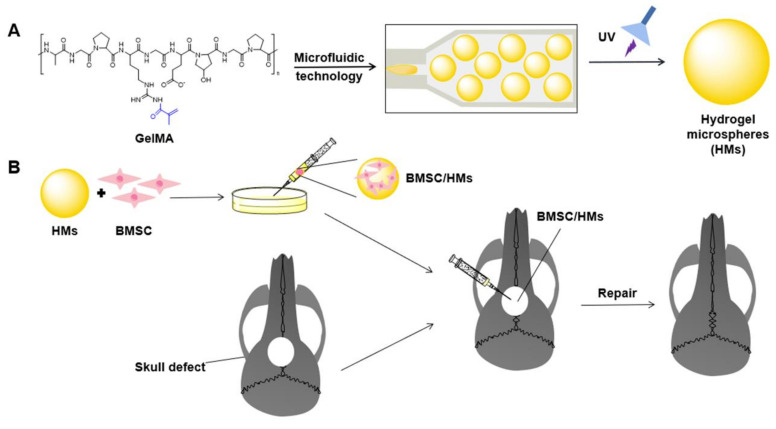
Schematic illustration of the preparation process of GelMA HMs and its treatment of skull defects. (**A**) HMs were fabricated by microfluidic technology with UV irradiation. (**B**) BMSCs were seeded on the HMs and then transplanted to the skull defects.

**Figure 2 gels-08-00275-f002:**
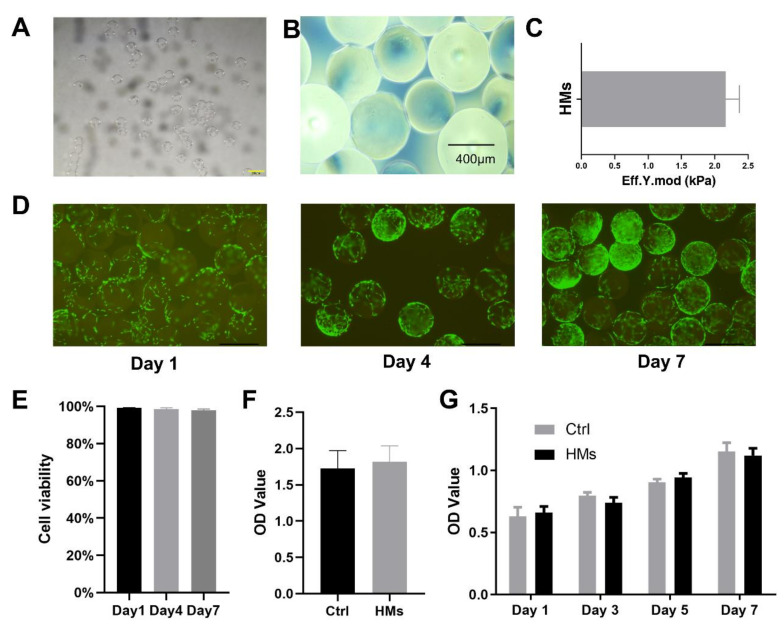
Physical and biocompatible characterization of HMs. (**A**) The stereoscopic microscope image of HMs. Scale bar: 500 μm. (**B**) The microscopic image of HMs. Scale bar: 400 μm. (**C**) The mechanical strength of HMs. (**D**) Live/dead staining of BMSCs co-cultured with HMs for 1, 4, and 7 days, respectively. Scale bar: 500 μm. (**E**) Viability quantification of BMSCs co-cultured with HMs for 1, 4, and 7 days. (**F**) The toxicity test of HMs and BMSCs co-culture. (**G**) The cytotoxicity of HMs to BMSCs after culture for 1, 3, 5, and 7 days.

**Figure 3 gels-08-00275-f003:**
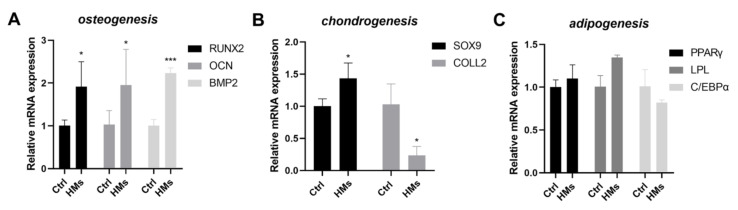
The effect of HMs on bone marrow mesenchymal stem cells. (**A**) The expression of osteogenic marker genes (RUNX2, OCN, and BMP2). (**B**) The expression of chondrogenic marker genes (SOX9 and COLL2). (**C**) The expression of adipogenic marker genes (PPARγ, LPL, and C/EBPα). (Error bars, mean ±SD; * *p* < 0.05, *** *p* < 0.001; *n* = 3 per group).

**Figure 4 gels-08-00275-f004:**
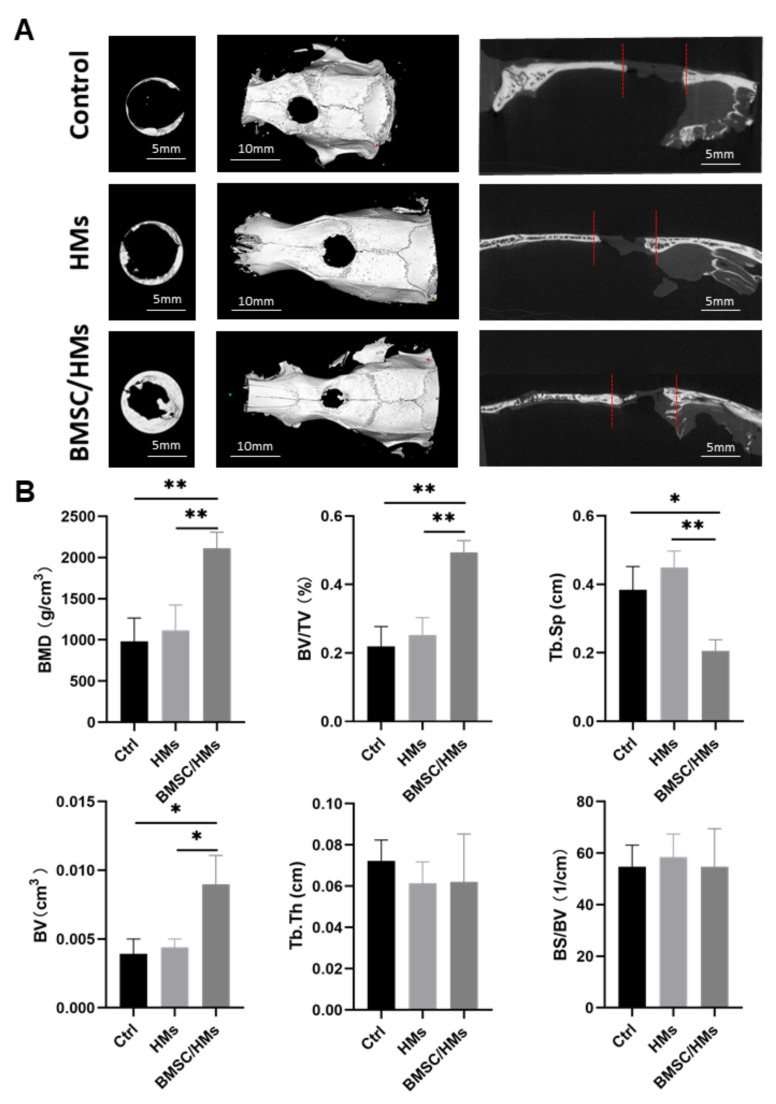
Evaluation of bone regeneration 8 weeks after surgery by micro-CT. (**A**) Micromorphometric analysis of large skull defects including superficial, three-dimensional, and sagittal views of micro-CT images after 8 weeks of surgery. (**B**) Micromorphometric bone parameters including BMD, BV/TV, Tb.Sp, BV, BS/BV, and Tb.Th analysis of control, HMs, and BMSC/HMs groups after 8 weeks of surgery (error bars, mean ± SD; * *p* < 0.05, ** *p* < 0.01; *n* = 3 per group).

**Figure 5 gels-08-00275-f005:**
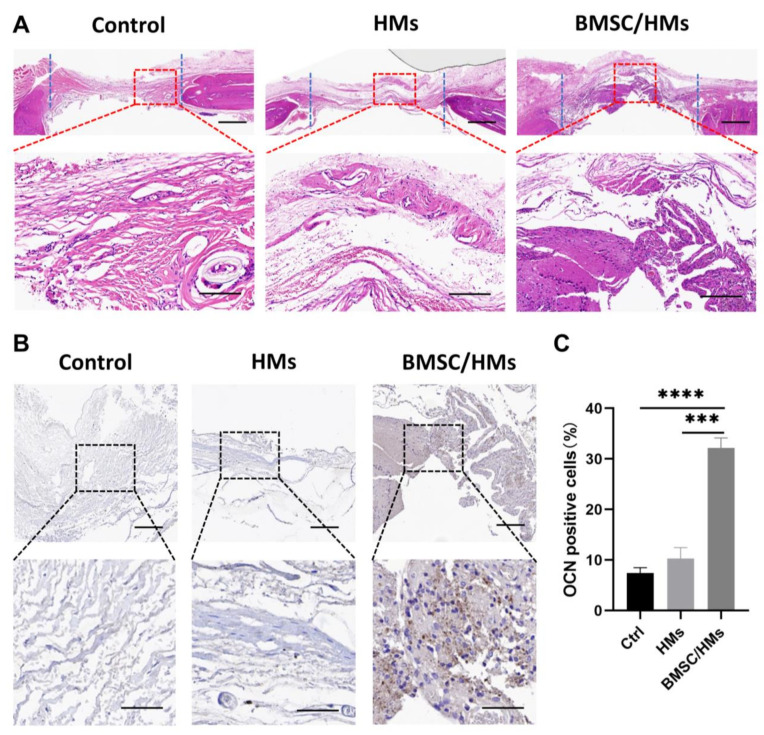
Histological analysis of bone regeneration 8 weeks after surgery. (**A**) Representative images for H&E staining in sections of control, HMs, and BMSC/HMs group for 8 weeks. Scale bar: 500 μm. The magnified images for HE staining in different groups in the area of skull defects for 8 weeks. Scale bar: 50 μm. (**B**) Representative images and magnified images for immunohistochemical staining of OCN-positive cells in the skull defect regions at 8 weeks post implantation in the three different groups. Scale bar of the first row: 50 μm. Scale bar of the second row: 20 μm. (**C**) Semi-quantitative analysis of the relative numbers of OCN-positive cells in the different groups (*n* = 3). *** *p* < 0.001, **** *p* < 0.0001.

**Table 1 gels-08-00275-t001:** Primers used for qRT-PCR.

Gene	Forward (5′-3′)	Reverse (5′-3′)
GAPDH	GGCAAGTTCAACGGCACAG	CGCCAGTAGACTCCACGACAT
RUNX2	AGAATGGACGTGCCCCCTA	CTGGGGAAGCAGCAACACTA
OCN	GCTCTGTGCTCCTGCATCTG	GCTCTGTGCTCCTGCATCTG
BMP2	CGCCTCACAAACAACCACAG	AATGACTCGGTTGGTCTCGG
SOX9	CTGACCGTGACCGTAGCAAGT	TGGATGTGGGCTTTGGACTCA
COLL2	GCTCCCAGAACATCACCTACCA	ATTCCTGCTCAGGCCCTCC
PPARγ	GAACGTGAAGCCCATCGAGGAC	GGAGCACCTTGGCGAACAGC
LPL	GCTGGCGTGGCAGGAAGTC	AGGCGACTAGGGGCTTCTGC
C/EBPα	CAAGAACAGCAACGAGTACCG	GTCACTGGTCAACTCCAGCAC

## Data Availability

Data are available from the authors.

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
