# Peer review of "Mesenchymal Stem Cells–Hydrogel Microspheres System for Bone Regeneration in Calvarial Defects"

_gels, 2022, doi:10.3390/gels8050275_

Round 1

Reviewer 1 Report

The manuscript showed that hydrogel microspheres promote osteogenic differentiation in vitro, and it combined with BMSCs can promote bone formation in the model of Cranial Bone Defect. However, the manuscript lacked rigorous expression about the description of many crucial data.
1. There is a lack of positive control groups like traditional hydrogels for repairing bone defects, so the proof of the therapeutic effect and superiority of HMs is insufficient, especially for the mechanism of hydrogel microspheres promoting osteogenic differentiation.
2. Figure 2 lacks the description of the G graph.
3. Method used to cultivate BMSCs in the CTRL group should be explained in results 2.1 and 2.2.   
4. The layout of the article needs to be modified.
5. High-resolution figures are required. like OCN in Fig5B cannot be distinguished.
6. The results showed that no bone regeneration was observed in the central area of the defect.Although the author had explained the result with weak mechanical strength. But it is not similar to the other results of hydrogel combined with BMSCs studies, Please explain this in more detail. 

Reviewer 2 Report

The semiquantitative methods used for analysis of histochemical and Immunohistochemical sections needs to be elaborated. 

Author Response

Point 1: The semiquantitative methods used for analysis of histochemical and Immunohistochemical sections needs to be elaborated.

Response 1: Thank you for this suggestion. We have added detailed methods of semi-quantitative analysis to 5.9. Immunohistochemistry (IHC).

“Osteocalcin (OCN) is a marker of new bone formation secreted by osteoblasts. Three sections of each group were randomly selected for OCN immunohistochemical staining. In simple terms, sections were incubated overnight with anti-OCN primary antibody (1: 100) at 4℃. Then, the secondary antibody (1: 500) was added and incubated at room temperature for 2 hours. After the staining was completed, light microscopy was used to observe the three randomly selected representative regions for semi-quantitative analysis of OCN positive staining. Image Pro Plus 6.0 software (Media Cybernetics) was used to measure the integral optical density (IOD) of OCN positive staining area. The intensity of each image was calculated by normalizing the IOD with the randomly selected areas of interest.”

Reviewer 3 Report

Dear authors

p4 in the legend of figure 2 : G instead of F

P7  Indicate in figure 4A the limits of the lesion with arrows

Figure 5 A To show bone regeneration it would be nice to do a more specific staining than HE like Gomori or alizarin red. It would be useful to do a calcein injection to show the newly formed bone.

Figure 5B Images are too small.

Implantation kinetics would make it possible to better see the formation of the bone, for example increasing the implantation time.

Best regards

Round 2

Reviewer 1 Report

The manuscript has been improved and it can be accepted after the issues of figures are resolved, like Fig5 is still low-resolution,  and all CT images lack of Scale-bar.

Author Response

Thanks for your suggestion.

  1. We have tried our best to improve the resolution of the picture and have made a replacement.
  2. Scale-bar of all CT images have been added.

Best regards!

Reviewer 3 Report

Dear authors

Thank you for making the corrections I requested. This study can be published in this form in gels.
Best regards

Author Response

We are glad to receive your reply. Thank you very much!

Best regards!